# Dissecting the Regulatory Network of Leaf Premature Senescence in Maize (*Zea mays* L.) Using Transcriptome Analysis of *ZmELS5* Mutant

**DOI:** 10.3390/genes10110944

**Published:** 2019-11-19

**Authors:** Mao Chai, Zhanyong Guo, Xia Shi, Yingbo Li, Jihua Tang, Zhanhui Zhang

**Affiliations:** National Key Laboratory of Wheat and Maize Crop Science, Henan Agricultural University, College of Agronomy, Zhengzhou 450046, China; chaimol@163.com (M.C.); gu0zhy@163.com (Z.G.); mrshi0614@126.com (X.S.); liybo2018@163.com (Y.L.); zhanhuiz15@icloud.com (Z.Z.)

**Keywords:** maize (*Zea mays* L.), leaf premature senescence, transcriptome analysis, regulatory networks

## Abstract

Leaf premature senescence largely determines maize (*Zea mays* L.) grain yield and quality. A natural recessive premature-senescence mutant was selected from the breeding population, and near-isogenic lines were constructed using Jing24 as the recurrent parent. In the near-isogenic lines, the dominant homozygous material was wild-type (WT), and the recessive material of early leaf senescence was the premature-senescence-type *ZmELS5*. To identify major genes and regulatory mechanisms involved in leaf senescence, a transcriptome analysis of the *ZmELS5* and WT near-isogenic lines (NILs) was performed. A total of 8796 differentially expressed transcripts were identified between *ZmELS5* and WT, including 3811 up-regulated and 4985 down-regulated transcripts. By combining gene ontology, Kyoto Encyclopedia of Genes and Genomes, gene set, and transcription factor enrichment analyses, key differentially expressed genes were screened. The senescence regulatory network was predicted based on these key differentially expressed genes, which indicated that the senescence process is mainly regulated by bHLH, WRKY, and AP2/EREBP family transcription factors, leading to the accumulations of jasmonic acid and ethylene. This causes stress responses and reductions in the chlorophyll a/b-binding protein activity level. Then, decreased ATP synthase activity leads to increased photosystem II photodamage, ultimately leading to leaf senescence.

## 1. Introduction

Plant senescence is the last stage of a developmental program, and it usually starts with leaf yellowing [1]. Leaf senescence determines crop grain yield and biomass formation, which is a highly regulated, well-coordinated, and biologically active process that marks the end of the life cycle of the leaf and, ultimately, the whole plant [2,3]. Delaying plant senescence can effectively prolong photosynthesis and increase the overall crop biomass [4,5,6,7,8]. However, premature senescence causes substantial production losses [9,10]. Leaf senescence is a complex physiological process that comprises of chlorophyll decomposition, photosynthesis termination, protein and nucleic acid degradation, molecular metabolism and nutrient transport decreases, and responses to cell death [11,12,13]. Many kinds of environmental factors can promote leaf senescence, such as drought, nutrient starvation, inhibited pollination, salinity stress, darkness, excessive light intensity, and biotic stresses [14,15,16,17]. The partitioning of sugars to various sinks and the associated signals emerging from the interplay of the photosynthetic output of the plant (source strength), as well as the photoassimilate (sink strength) needs, play key roles in the regulation of senescence [16,18,19,20]. Exploring the regulatory mechanisms underlying plant senescence is essential for crop improvement. However, the corresponding mechanisms in maize are still confusing, especially the onset and precise controlling of the senescence progress.

In recent years, some senescence-associated genes (SAGs) have been identified in various plant species at the transcriptional level [21,22,23]. For instance, almost one fourth of *Arabidopsis* genes are identified to be associated with senescence, as assessed by transcriptome analyses [24]. Numerous studies have identified many key regulatory molecules, as well as a series of signaling pathways, that are involved in gene expression changes during leaf senescence [25]. These regulatory factors include chromatin-modifying factors (DDM1, DRD1, and HDA9-PWR), transcription factors (NAC, bHLH, JAZ, MYC, and WRKY), post-transcriptional regulatory factors, post-translational regulators (MAPK6, MAPKKK18, ATL31, PUB12/13, PUB44, and SSPP), and metabolic regulators [26,27,28]. Importantly, over 150 *Arabidopsis* mutants and/or transgenic plants have altered leaf senescence phenotypes [29].

Maize is an important crop for human food and animal feed, as well as an important genetic model plant. Anthesis represents the onset of plant senescence, during which maize leaves start to turn to yellow and gradually die. As leaves senesce, their photosynthetic rates decline. Thus, manipulating the maize leaf senescence process could help achieve high grain yield and quality. In the United States of America (USA) and Canada, maize yield was significantly increased by delaying leaf senescence [30,31]. Similarly, owing to delayed leaf senescence, rice grain yields increased by 24% [32] and 10.3% [33,34], respectively. Recently, several omics analyses have been performed to identify senescence-associated genes [35,36,37,38], miRNAs [5], and proteins [39,40] in maize. Although large amounts of genes have been screened, only *ZmSnRK1*s and *knotted1* were experimentally verified [41,42].

Because of the complicated regulatory networks, it is difficult to identify major senescence-associated genes. Premature senescence mutants provide ideal materials for functionally identifying major senescence-associated genes and corresponding regulatory networks. In our previous study, a premature-senescence mutant, *early leaf senescence 5* (*ZmELS5*), was screened from breeding population. In the present study, we aimed to identify senescence-associated genes using a comparative transcriptome analysis between *ZmELS5* and wild type (WT) NILs; to construct the corresponding regulatory networks; and to predict a regulatory model for *ZmELS5* in maize leaf senescence.

## 2. Materials and Methods

### 2.1. Plant Materials

In our previous study, the premature senescence-mutant *ZmELS5* in maize was screened from a breeding population and crossed with the normal Jing24 to create near-isogenic lines (NILs) through five back-crosses. In the NILs, the dominant homozygous material was WT, and the recessive material responsible for the early leaf senescence was the premature-senescence type *ZmELS5*. In the summer of 2016, the *ZmELS5* NILs were planted in the Science and Education Park of Henan Agricultural University (Zhengzhou, China; E113°35′, N34°51′). In the field, the leaves of *ZmELS5* NILs started to yellow at V6 stage. At V13 stage, the 11th leaf (ear leaf) of mutants and WT were immediately sampled, frozen in liquid nitrogen, with three biological replicates, and further stored at −80 °C in a refrigerator. Additionally, the candidate gene of *ZmELS5* was mapped in a 9.58 Mb chromosomal region at Bin 8.03.

### 2.2. RNA Extraction

Samples of the 11th leaves of the *ZmELS5* and its WT NILs were used to extract total RNA. Total RNAs were extracted using TRIzol reagent (Invitrogen, Waltham, MA, USA) following the manufacturer’s instruction. Six libraries were constructed and sequenced using the Illumina HiSeq 2500 platform (Hangzhou Lianchuan Biotechnology Co. Ltd., Hangzhou, China). The entire original sequence data in fastq format have been uploaded to the NCBI short read archive (accession number: PRJNA555720).

### 2.3. Transcriptome Analysis

To screen for SAGs, a comparative transcriptome analysis was conducted within *ZmELS5* and its WT NILs. The raw data of RNA sequencing (RNA-seq) were quality controlled using FastQC (http://www.bioinformatics.babraham.ac.uk/projects/fastqc/). The Q30 ratios of the six sequenced libraries were greater than 93%. The trimmed and low quality (Q < 30) sequencing data were removed by Trimmomatic Software V0.36 [43], and 34.26 Gb sequence data were acquired (Appendix A). Then, the clean sequencing data were aligned to the maize B73 RefGen_V4.42 reference genome (http://ensembl.gramene.org/Zea_mays/Info/Index) using HISAT2 V2.1.0 [44]. Sorting and converting the sam file to a bam file was done using SAMtools software V1.9 [45]. The transcripts were assembled using StringTie software V1.3.4 [46], and the count matrix was generated using prepDE.py (https://ccb.jhu.edu/software/stringtie/dl/prepDE.py). The differentially expressed gene (DEG) analysis was performed using DESeq2 software V1.22.2 [47]. A principal component analysis (PCA) was performed using the DESeq2 VST mode. Transcripts with a log_2_ fold change >2 and a false discovery rate-corrected *p*-value <0.05 were determined to be differentially expressed. We used the apeglm method for LFC effect-size shrinkage [48], which improved on the previous estimator.

The gene ontology (GO) and Kyoto Encyclopedia of Genes and Genomes (KEGG) enrichment analyses were performed using the maize profile database (org. Zeamays; e.g., sqlite) in the clusterProfiler software V3.10.1 [49], and AnnotationHub (V2.14.5) R package [50]. The enrichment analyses used the ensemble database to convert the gene number (maizegbdId) to the corresponding Entrez ID.

The gene set enrichment analysis (GSEA) analysis used Cluster Profiler’s [51] gseGO command. GSEA uses all the transcript data sets expressed in the sequenced materials. The FTP database based on the Ensembl genome (ftp://ftp.ensemblgenomes.org/pub/current/plants/tsv/zea_mays/) was used to convert transcription IDs into Entrez IDs using a Python 3 script. For the GSEA, gseGO parameters were set as repeat 1000 times, nPerm = 1000, minGSSize = 15, and maxGSSize = 500 [52]. In addition, the online tool MCENet (http://bioinformatics.cau.edu.cn/MCENet/index.php) was used for the transcription factor enrichment analysis of the gene sets [53]. Based on the transcription factors corresponding to the expressed genes, the transcription factors corresponding to the DEGs were selected for the enrichment analysis.

### 2.4. Quantitative Reverse Transcription PCR (RT-qPCR) Validation

The gene expressions, which correlated with premature senescence, were verified by RT-qPCR. RT-qPCR primers (http://primer3.ut.ee/) are listed in Appendix A. The relative expression levels of DEGs identified in the transcriptome analyses were measured using the PrimeScript™ RT reagent kit with gDNA Eraser (Perfect Real Time) and the SYBR^®^ Premix EX Taq™ II (Tli RNaseH Plus) Kit (TaKaRa, Dalian, China). The RT-qPCR was performed using the CFX96 Touch™ Real-Time PCR Detection System (Bio-Rad, Hercules, CA, USA). The *Actin* gene was used as the endogenous control [54]. The relative expressions of RNAs and targets were calculated using the 2^−ΔΔCt^ method [55].

### 2.5. Determination of Chlorophyll Content

In total, 0.1 g leaf was soaked in 80% acetone for 24 h. The chlorophyll content was determined using a spectrophotometer (UV-3200PC, Shanghai MAPADA Instruments Co. Ltd., Shanghai, China) according to the method by Porra [56]. Three biological replicates were performed per material, with three technical replicates per biological replicate.

### 2.6. Putative Senescence Regulatory Network

The senescence regulatory network was predicted using the online tool STRING V11 (https://string-db.org/) [57] by the GO and GSEA enrichment analyses for these identified important DEGs.

## 3. Results

### 3.1. Characterization of Maize Premature-Senescence Mutant ZmELS5

From 2016 to 2019, *ZmELS5* and its WT NILs were planted in different environments (Zhengzhou, China, E113°35′, N34°51′; Sanya, China, E108°92′, N18°44′). In the field, the *ZmELS5* NIL plants began to senesce at the V6 stage, as indicated by their leaves gradually turning yellow from the bottom up, and by the V13 stage, the 12th leaves margin of the *ZmELS5*’s started to yellow (Figure 1A). Furthermore, the 11th leaves (ear leaves) of *ZmELS5* NIL plants showed significant senescence phenotypes (Figure 1B). However, all the leaves of the WT NIL plants were green. The 11th leaves (ear leaves) were sampled as shown in Figure 1B, and the chlorophyll content was measured as shown in Figure 2. During this period, the chlorophyll content in *ZmELS5* was significantly lower than that in WT. Chlorophyll a and chlorophyll b were significantly lower in mutants than in WT. This indicates that, at this stage, the mutant has started to senesce. There were significant differences in the metabolic activities between the *ZmELS5* and WT NILs.

### 3.2. Differentially Expressed Gene Identification

For the comparative transcriptome analysis, six samples (three replicates each for the WT and *ZmELS5* NILs) were used to construct cDNA libraries and for RNA-seq, and 34.26 Gb raw data were obtained. After low quality reads were filtered, the clean reads were then aligned with the maize B73 genome (RefGen_V4.42). Approximately 85% of the clean reads were unique and could be mapped (Table 1).

Between WT and *ZmELS5* NILs, 8796 (log_2_ fold change >2.0, count >10, 10.2% of 85,861 total) significant differentially expressed transcripts were identified, including 4985 down-regulated transcripts and 3811 up-regulated transcripts. Among them, 5428 differentially expressed transcripts were annotated, 2292 transcripts (1808 genes) were up-regulated, and 3136 transcripts (2746 genes) were down-regulated (Figure 3). Of the DEGs, the up-regulated genes *Zm00001d016441*, *Zm00001d036370*, *Zm00001d024767*, *Zm00001d045251*, and *Zm00001d014993*, and the down-regulated genes *Zm00001d048702*, *Zm00001d010821*, *Zm00001d026405*, *Zm00001d018460*, and *Zm00001d004855*, showed the most significant differences. These genes may play important roles in *ZmELS5* NILs’ premature leaf senescence.

Based on the Annotation Hub database, these DEGs were enriched to different classes and further analyzed. In the GO analysis, the identified DEGs were classified into three classes: Molecular function, biological process, and cellular component-related genes (Figure 4A). The molecular function-related genes were all enriched in the chlorophyll-binding group. Most of the biological process-related genes were involved in photosynthesis, precursor metabolite generation, and abiotic stimulus responses (Figure 4A and Figure 5, and Appendix A). The other DEGs were mainly enriched in the synthesis of thylakoid, thylakoid part, thylakoid membrane, chloroplast, photosystem (PS) I, and photosynthetic membrane (Figure 4A and Figure 6).

The KEGG analysis indicated that the DEGs were mainly enriched in carbon metabolism, starch and sucrose, amino sugar and nucleotide sugar metabolism, protein processing in endoplasmic reticulum, photosynthesis, and photosynthesis-antenna proteins (Figure 4B).

The GSEA was performed on all the tested genes in WT and *ZmELS5* NILs (Figure 7 and Figure 8). The genes up-regulated during the leaf senescence of *ZmELS5* NILs were mainly distributed in cell membranes, vacuoles, and other cell components, and mainly involved in the regulation of stress response. The down-regulated genes were mainly distributed in thylakoid, chloroplast, light system, and other related cell components, and they were mainly involved in the regulation of photosynthesis and light response (Figure 7). The enrichment of GSEA cell components revealed two core networks. One is the network of photosynthesis and the other is the vacuole-related network (Figure 8). The GSEA enrichment results indicated that most of the up-regulated genes were located in vacuoles and tonoplasts. There were 16 genes co-expressed in the vacuolar core module of the GSEA co-expression analysis. The functions of the genes enriched in the vacuoles mainly included histone H4, β-galactosidase 15, hydrogen exchanger 4, and tonoplast intrinsic protein 3.

There were 14 co-expressed genes in the light and action core modules of GO and GSEA (Table 2). The functions of these genes were mainly involved with the light-harvesting complex, which consists of chlorophylls a and b and the chlorophyll a/b-binding protein. These genes were significantly down-regulated in the *ZmELS5* NILs.

### 3.3. Characterizing the Expression of Transcription Factor Genes in Leaf Senescence

Transcription factors play critical roles in the onset of leaf senescence. Of the identified 3170 DEGs during maize leaves senescence, 170 transcription factors were identified that belonged to 17 transcription factor families, including WRKY, bHLH, MYB, NAC, and ERF (Appendix A). In particular, the WRKY, NAC, MYB, bHLH, and HD-ZIP transcription factor families had significantly differential expressions that were induced by premature senescence. Most of these transcription factor families have been identified as important leaf senescence regulators in *Arabidopsis*.

### 3.4. Verifying Differential Gene Expression Profiles Using RT-qPCR

DEGs involved in multiple core pathways during leaf senescence were selected for RT-qPCR validation. These genes mainly encoded chlorophyll a/b-binding protein, light harvesting complex, PSI subunit O, heat shock proteins, and WRKY transcription factors. The genes were significantly differentially expressed between WT and *ZmELS5* NILs, as assessed by RT-qPCR (Figure 9). Of them, chlorophyll a/b-binding protein, light harvesting complex a/b-binding protein, and PSI subunit O-related genes were significantly down-regulated, while heat-shock protein and WRKY transcription factor-related genes were up-regulated. The RT-qPCR results also revealed that the expression levels of the selected genes were consistent with those determined using RNA-seq, and the two showed a strong correlation. The expression pattern obtained by RT-qPCR was strongly correlated with the RNA-seq results (*R^2^* = 0.8981), indicating that the RNA-seq data are reliable (Appendix A).

## 4. Discussion

### 4.1. Key Genes in ZmELS5 Leaf Premature Senescence

Leaf senescence is accompanied by a change in leaf color, which reflects chlorophyll loss, and ends with the death or abscission of the leaf. In the cells of senescing leaves, a highly ordered disassembly and degradation of cellular components occurs [58]. During leaf senescence, the chlorophyll is gradually degraded with the thylakoid membrane converting to a colorless decomposition product that is stored in the vacuole through a multi-step pathway. The order in which the various components in the leaves senesce is as follows: Chlorophyll, chlorophyll-binding protein, PSII, and PSI. In the present study, the leaves of *ZmELS5* NILs gradually senesced from the bottom. DEGs were identified using a comparative transcriptome analysis. The results revealed that the genes involved in photosynthesis-related and stress responses, and the cellular components of the thylakoid, thylakoid membrane, chloroplast, PSI, and photosynthetic membrane, were all down-regulated, which is consistent with chlorophyll decomposition and decreasing photosynthesis. However, the genes mainly encoding protein degradation-related enzymes, transcription factors, and stress-responsive proteins were up-regulated. They may be involved in the onset of leaf senescence and senescence-induced metabolisms. Of them, the homologue of *Zm00001d016441* in *Arabidopsis* encodes a kind of tyrosine aminotransferase that is involved in stress tolerance [59]. The gene *Zm00001d036370* encodes chitinase, which is expressed in plants in response to abiotic stresses and during developmental processes [60]. The gene *Zm00001d024767* encodes aspartic proteinase A1, and its homologue in *Arabidopsis* is involved in several physiological processes in plants, including protein processing, senescence, and stress responses [61]. The gene *Zm00001d014993* encodes luminal-binding protein 2 that has a role in the regulation of tobacco stress-induced programmed cell death [62]. The gene *Zm00001d049217* is a homologue of *Arabidopsis SAG12*, which regulates senescence processes in a Gibberellic acid-dependent pathway [63]. The genes *Zm00001d043025*, *Zm00001d003412*, and *Zm00001d012427* encode WRKY33, P-type R2R3 Myb protein, and ZAT11 transcription factor, respectively, which respond to abiotic stresses in *Arabidopsis* [64,65,66]. Additionally, several pathogenesis-related genes were significantly up-regulated, such as *Zm00001d009296* (*ZmPRms*), *Zm00001d018738* (*prp4*), and *Zm00001d023811* (*PR10*). In maize, *ZmPRms* possibly to play an important role in the defense of aflatoxin infection [67]; *Prp4* is involved in responding to the infection by *Aspergillus flavus* [68], and *PR1* family genes were associated with *Fusarium graminearum* infection [69], which is the primary pathogen causing stalk rot in maize [70]. This indicates that pathogenesis-related genes (PR proteins) play crucial roles in plants’ defense system and leaf senescence.

In our previous study, the candidate gene of *ZmELS5* mutant senescence was mapped to the chromosomal region Chr8: 86,964,414-92,548,025. In this region, the several differently expressed genes were distributed, such as *Zm00001d009945*, *Zm00001d009910*, *Zm00001d009903*, *Zm00001d009928*, and *Zm00001d009936*. *Zm00001d009910* encoded a NADH protein. *Zm00001d009928* encoding photosystem II reaction center PsbP family protein was highly down-regulated. *Zm00001d00009936* encoded a Basic endochitinase A protein.

### 4.2. Transcription Factors Involved in the Regulation of ZmELS5 Leaf Senescence

Transcription factors play critical roles in responding to different environmental factors and to the onset of many biological progresses. To date, several transcription factor families have been identified as taking part in the regulation of leaf senescence progress, such as NAC, WRKY, MYB, C2H2 zinc-finger, bZIP, and AP2/EREBP [71]. In *Arabidopsis*, the WRKY family of transcription factors plays an important role in regulating plant leaf senescence by a jasmonic acid (JA)-dependent pathway or chrlorophyll degradation [17,72,73,74], such as WRKY53 [74] and WRKY6 [75]. In global transcriptome profiling, more than 30 *NAC* genes showed enhanced expression levels during natural leaf senescence in *Arabidopsis*, highlighting their importance in plant senescence regulation [21,76], such as ORESARA1 [77] and Oresara1 sister 1 (ORS1) [78]. The bHLH subgroup IIId factors (bHLH03, bHLH13, bHLH14, and bHLH17) bind to the promoter of *SAG29* and repress its expression to attenuate MYC2/MYC3/MYC4/MYC5-activated JA-induced leaf senescence [79,80]. The *Arabidopsis* MYBS3 plays a critical role in the developmentally-regulated and dark-induced leaf senescence by repressing transcription [81]. In addition, heat shock proteins are involved in the degradation of protein components during leaf senescence [82], such as heat shock protein (Hsp) 70 and Hsp90. In maize, several TF families have been identified to be associated with leaf senescence by RNA-seq profiling, such as NAC, MYB, bHLH, AP2/EREBP, C2h2, and bZIP families [83]. However, only NAC007 has been experimentally verified [38]. In the transcription factor enrichment analysis of the present study, these transcription factor family members were determined to be associated with the premature leaf senescence of *ZmELS5* NILs. Transcription factors, such as WRKY, TCP, MYC, and NAC, usually take part in leaf senescence regulation through a JA signaling-dependent pathway. Chlorophyll breakdown is a feature of JA-induced, as well as natural, leaf senescence, and the JA-responsive enzyme chlorophyllase is the main enzyme responsible for chlorophyll breakdown [84].

### 4.3. Predicted Regulatory Networks of Maize Leaf Senescence

The onset of leaf senescence depends on developmental stages and environmental signals. Once leaf senescence begins, concerted cellular events are triggered, such as rapid chlorophyll degradation, increased membrane ion leakage, and, eventually, the decline of photochemical biosynthesis [85]. We predicted the possible senescence regulatory networks using important DEGs and transcription factors (Figure 10 and Figure 11). The premature senescence of the *ZmELS5* NILs may be regulated by a variety of signals, ultimately leading to leaf senescence. In the JA-regulated senescence signaling pathway in *Arabidopsis*, the bHLH transcription factor negatively regulates the senescence gene *SAG29* [86]. Signal pathways for leaf senescence and plant defense responses may overlap [14,87]. The upstream bHLH86 transcription factor of the senescence regulatory network negatively regulates stress-responsive proteins. The homeodomain-leucine zipper transcription factor hb12 up-regulates expression during senescence initiation and positively regulates stress-responsive proteins. Stress-responsive proteins, such as Hsp90, luminal-binding protein, and chitinase 2, are up-regulated. The stress-responsive proteins positively regulate the chlorophyll a/b-binding proteins. The expression levels of chlorophyll a/b-binding protein, as well as *Cab48*, *Lhcb3*, *gpm571*, and *Zm00001d011285*, gradually decrease as the senescence process extends. The corresponding chlorophyll content gradually decreases, leading to the senescence of phenotypic leaves. The *ndh* gene is involved in the protection of chloroplasts from photooxidative stress in mature senescent leaves [88]. The down-regulated expression of *ndhE* and *ndhK* reduces the inhibition of photooxidative stress on chloroplasts in senescent leaves and aggravates the degradation of chloroplast proteins. The decreased ATP synthase activity leads to increased PSII photodamage [89]. Starch synthase4 is expressed in a specific region of the thylakoid and restricts the formation of starch granules in these areas of the chloroplast. The down-regulation of starch synthase4 leads to the accumulation of polymers as large as starch granules in this region of the thylakoid, which affects photosynthetic reactions [90], ultimately leading to leaf senescence. The WRKY53 transcription factor activates *SAG12* from upstream. This eventually leads to the accumulation of JA, a decreased chlorophyll content, decreased PSII activity, and the promotion of senescence [17]. The accumulation of JA causes a stress response. The overexpression of ERF1 induces defense responses in ethylene or JA mutants, as well as an induction of many other ethylene- and JA-reactive genes [91].

### 4.4. SAG Mutant Identification and the Dissection of Related Regulatory Mechanisms

Leaf senescence is a complex process that is regulated by intricate gene networks in plants [25,92]. Uncovering the related gene regulatory networks is critical for crop improvement. A comparative omics analysis and bioinformatics analysis will be helpful for *SAG* screening and constructing the related regulatory networks. However, the SAG functions still need to be experimentally verified, as well as the identities of key SAGs. Leaf senescence-related mutants can provide materials for identifying key *SAG*s and the corresponding regulatory mechanisms. In the present work, we conducted a comparative transcriptome analysis between the premature-senescence mutant *ZmELS5* and its WT NILs. This transcriptome analysis found important DEGs and pathways involved in the process of premature leaf senescence, and, in combination with previous studies, can aid in predicting possible regulatory networks for premature senescence. This provides a theoretical basis for further gene cloning and functional validation.

## Figures and Tables

**Figure 1 genes-10-00944-f001:**
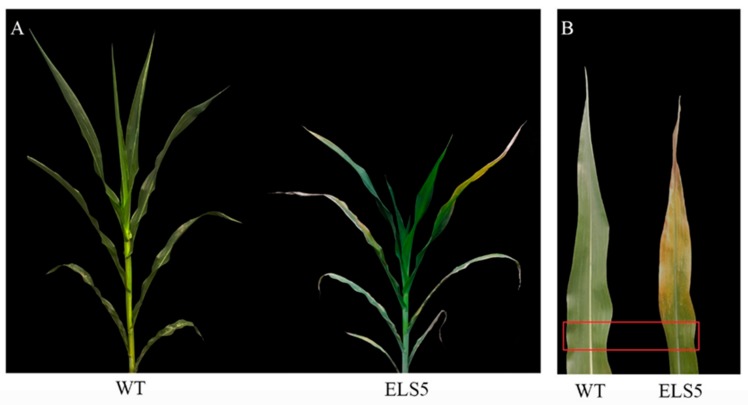
Phenotyping of wild-type (WT) and *ZmELS5* near-isogenic lines (NILs) at V13 stage: (**A**) Whole plant of WT and *ZmELS5* NILs at V13 stage; (**B**) the 11th leaf phenotype of the WT and *ZmELS5* NILs at V13 stage.

**Figure 2 genes-10-00944-f002:**
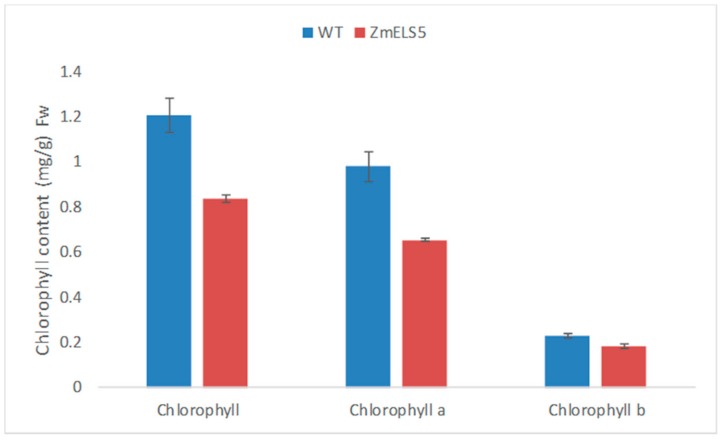
Chlorophyll contents in the leaves of WT and *ZmELS5* NILs.

**Figure 3 genes-10-00944-f003:**
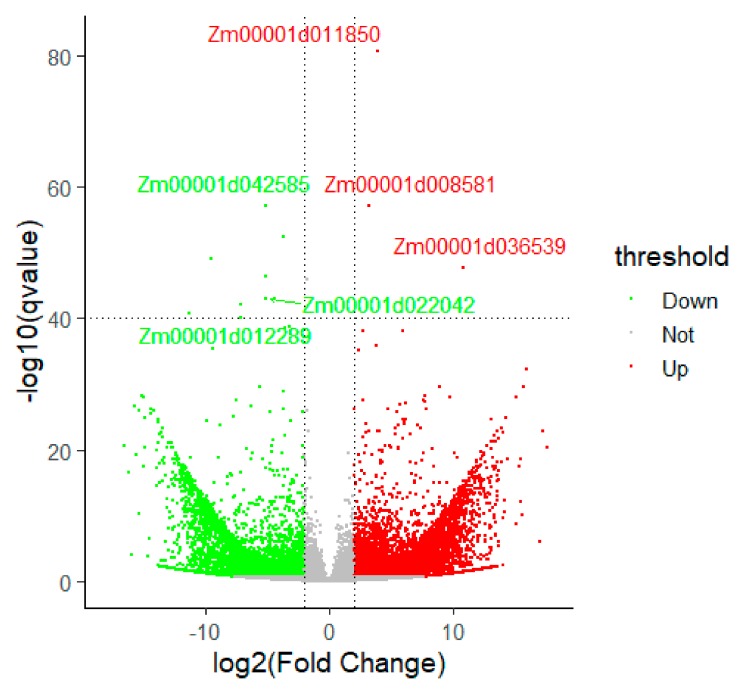
Functional classification of differentially expressed genes. Red dots indicate up-regulated genes and green dots indicate down-regulated genes under significant levels of Padj <0.05 and Log_2_ fold change >2.

**Figure 4 genes-10-00944-f004:**
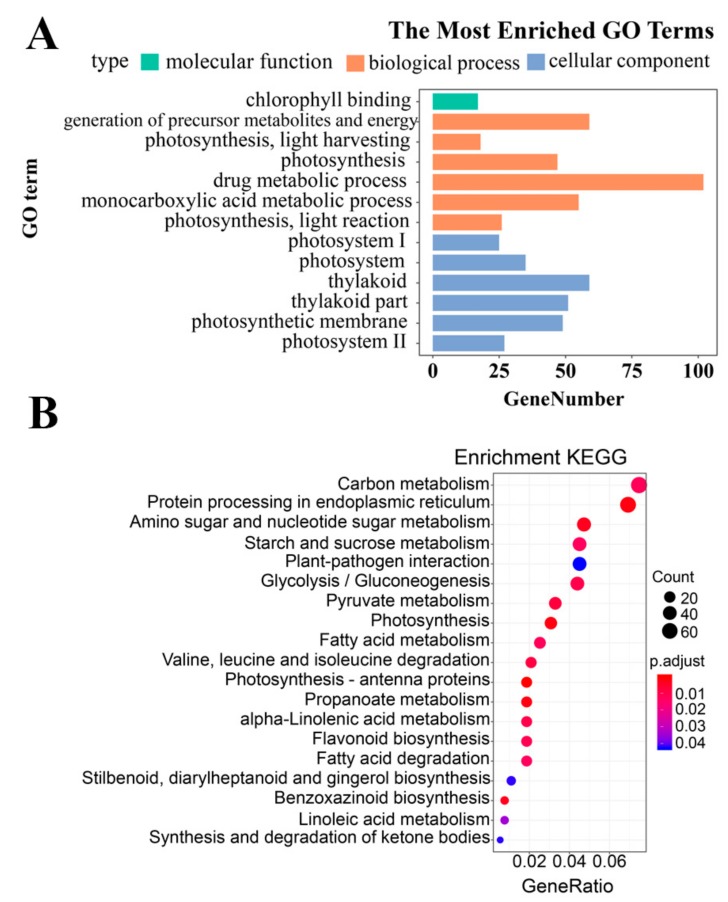
(**A**) Gene ontology (GO) and (**B**) Kyoto Encyclopedia of Genes and Genomes (KEGG) enrichment analysis of differentially expressed genes.

**Figure 5 genes-10-00944-f005:**
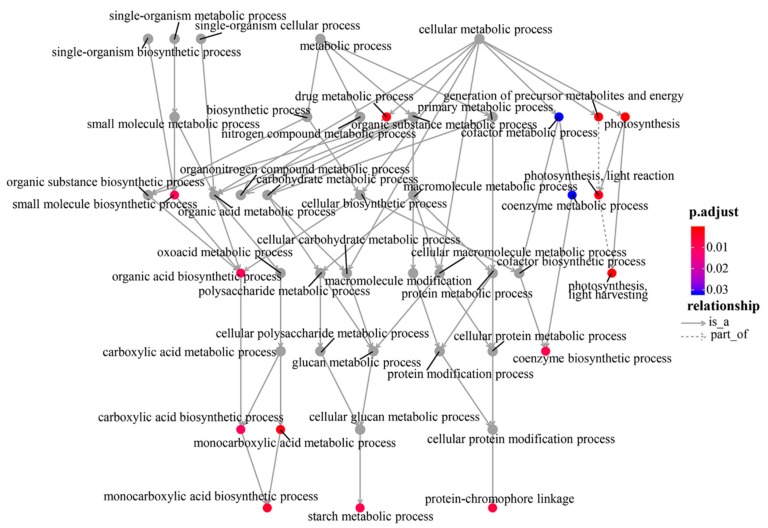
Gene and biological process connect network.

**Figure 6 genes-10-00944-f006:**
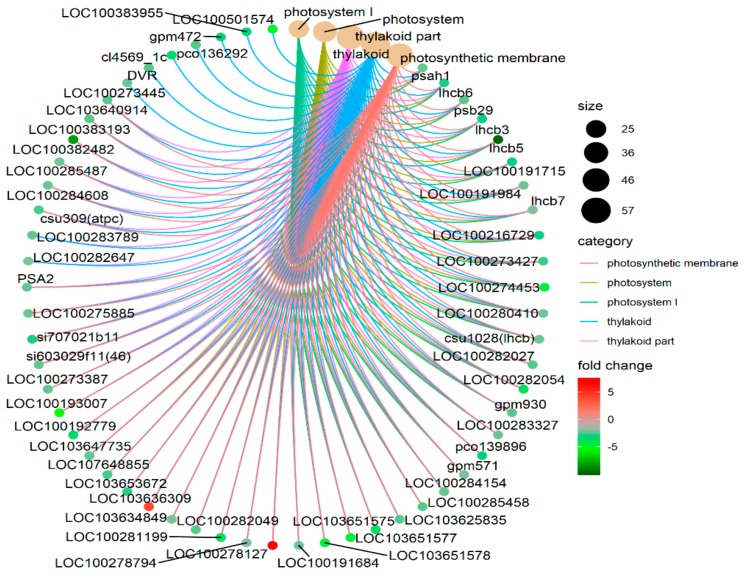
Different cellular components connect network.

**Figure 7 genes-10-00944-f007:**
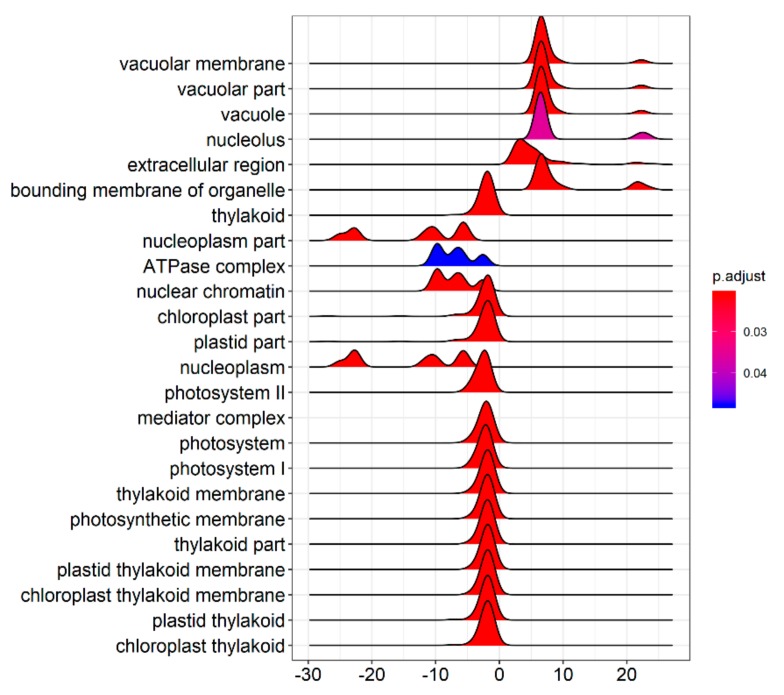
Gene set enrichment analysis of cell component distribution.

**Figure 8 genes-10-00944-f008:**
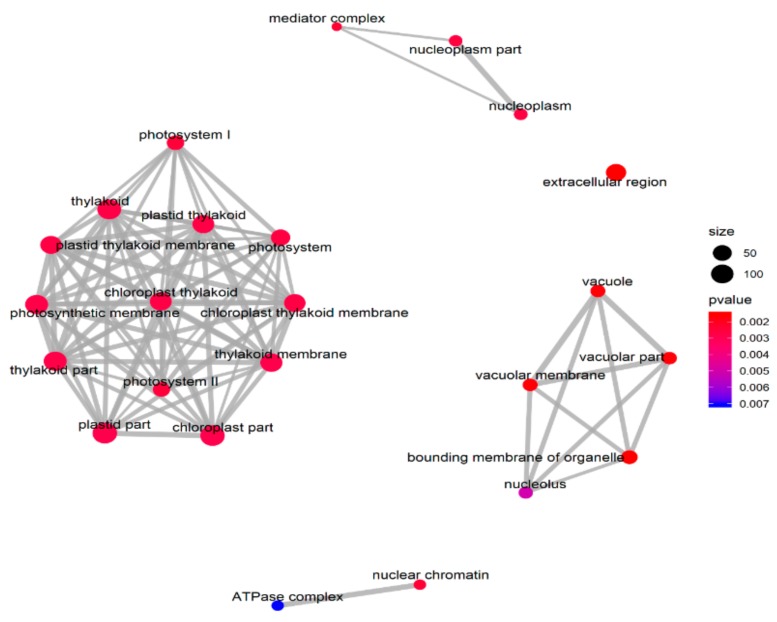
Network of roles of cellular components enriched in gene sets.

**Figure 9 genes-10-00944-f009:**
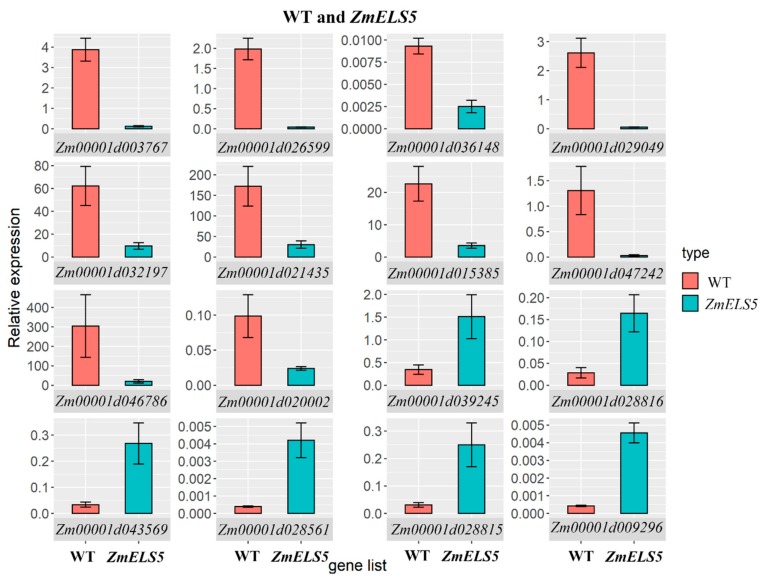
Comparison of expression of WT and *ZmELS5* NILs.

**Figure 10 genes-10-00944-f010:**
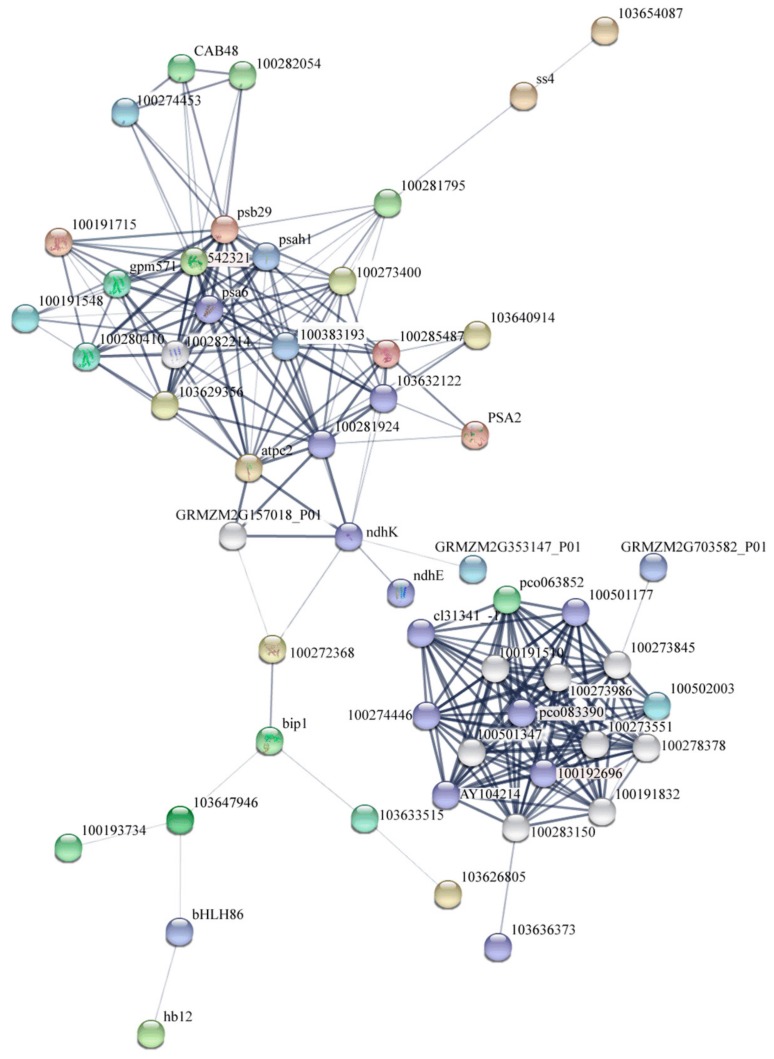
The string network of proteins (encoded by the differentially expressed genes (DEGs)) involved in senescence signaling pathways.

**Figure 11 genes-10-00944-f011:**
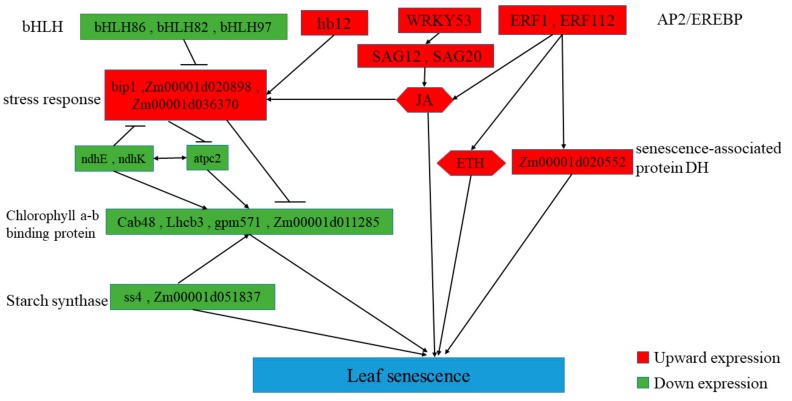
Predicting the regulatory process of premature senescence of mutant leaves.

**Table 1 genes-10-00944-t001:** Overview of mapping quality in RNA-seq data.

Sample	Raw Data Reads	Reads After Filter	Clean Reads Rate (%)	Mapped Reads	Mapped Rate (%)	Unique Mapped Reads	Unique Mapped Rate (%)
*ZmELS 5-1*	40,957,988	39,400,718	96.20	35,598,549	90.35	33,880,944	85.99
*ZmELS 5-2*	40,996,812	39,317,518	95.90	35,749,713	91.18	34,040,734	86.58
*ZmELS 5-3*	42,892,702	41,233,326	96.13	37,786,220	91.64	36,022,486	87.36
WT-1	42,805,416	41,244,366	96.35	37,421,013	90.73	35,435,962	85.92
WT-2	40,873,684	39,408,286	96.41	36,511,777	92.65	35,173,768	89.25
WT-3	42,401,032	40,813,916	96.26	37,254,943	91.28	35,383,646	86.70

**Table 2 genes-10-00944-t002:** Differentially expressed genes screened by GO analysis and gene set enrichment analysis (GSEA) in *ZmELS5* NILs leaf senescence.

Gene ID	BaseMean	Log_2_ Fold Change	*p*-Value	Homologous *Arabidopsis* Genes	Chr	Annotation of Maize	Source
*ZemaCp012*	13.17	−2	0.001891	ATCG00210	Pt	cytochrome b6/f complex subunit N	GSEA_CC
*Zm00001d039040*	170,043.95	−2.02	0.000026	AT2G34420	6	light harvesting complex mesophyll 7	GO_BP,GSEA_CC
*ZemaCp019*	1122.6	−2.03	0.000237	ATCG00130	Pt	ATPase subunit I	GSEA_CC
*Zm00001d032197*	39,958.88	−2.11	0.005955	AT3G47470	1	Chlorophyll a-b binding protein 4 chloroplastic	GO_BP,GSEA_CC
*Zm00001d032331*	1541.9	−2.15	8.35 × 10^−8^	AT3G01480	1	peptidyl-prolyl cis-trans isomerase	GSEA_CC
*ZemaCp041*	48.41	−2.18	0.000427	ATCG00590	Pt	cytochrome b6/f complex subunit VI	GSEA_CC
*Zm00001d050403*	23,382.97	−2.19	0.006146	AT3G47470	4	chlorophyll a-b binding protein 4	GO_BP,GSEA_CC
*Zm00001d018157*	5238.44	−2.22	7.28 × 10^−7^	AT1G45474	5	light harvesting complex a/b protein 4	GO_BP,GSEA_CC
*Zm00001d020030*	522.23	−2.23	0.003533		7	putative NAD(P)H dehydrogenase subunit CRR3 chloroplastic	GSEA_CC
*ZemaCp042*	108.17	−2.24	0.002738	ATCG00600	Pt	cytochrome B6-F complex subunit 5	GSEA_CC
*ZemaCp043*	18.23	−2.27	0.000883	ATCG00630	Pt	photosystem I subunit IX	GSEA_CC
*ZemaCp088*	28.03	−2.28	0.002185	ATCG01070	Pt	NADH dehydrogenase subunit 4L	GSEA_CC
*Zm00001d038984*	57,330.44	−2.31	0.001118	AT3G16140	6	photosystem I H subunit 1	GSEA_CC
*Zm00001d030762*	19,126.21	−2.33	0.001473	AT5G08050	1		GSEA_CC
*Zm00001d040242*	401.87	−2.36	0.001072	AT1G71480	3	Nuclear transport factor 2 (NTF2) family protein	GSEA_CC
*Zm00001d031738*	2102.21	−2.38	0.005397	AT2G34860	1	photosystem I	GSEA_CC
*Zm00001d009877*	4724.03	−2.43	0.007421	AT3G46780	8	Protein plastid transcriptionally active 16 chloroplastic	GSEA_CC
*ZemaCp011*	119.68	−2.45	2.44 × 10^−8^	ATCG00220	Pt	photosystem II protein M	GSEA_CC
*Zm00001d021763*	119,242.53	−2.49	0.00516	AT2G40100	7	photosystem II subunit 29	GO_BP,GSEA_CC
*Zm00001d023757*	6515.79	−2.52	0.000234	AT5G21430	10	NAD(P)H-quinone oxidoreductase subunit U chloroplastic	GSEA_CC
*Zm00001d021620*	23,073.49	−2.61	8.91 × 10^−12^	AT4G04640	7	ATP synthase chloroplast subunit 2	GSEA_CC
*Zm00001d006663*	2511.32	−2.69	0.000695	AT3G61470	2	chlorophyll a-b binding protein	GO_BP,GSEA_CC
*Zm00001d026599*	22,411.85	−2.81	0.004233	AT1G15820	10	light harvesting chlorophyll a/b binding protein 6	GO_BP,GSEA_CC
*Zm00001d006587*	18,274.92	−2.87	0.004411	AT3G08940	2	Chlorophyll a-b binding protein CP29.1 chloroplastic	GO_BP,GSEA_CC
*Zm00001d009589*	26,091.22	−2.93	0.0013	AT2G34420	8	light harvesting chlorophyll a/b binding protein 3	GO_BP,GSEA_CC
*Zm00001d045620*	935.16	−3.4	0.007634		9	Plastocyanin major isoform chloroplastic	GSEA_CC
*Zm00001d044396*	137.84	−3.76	2.25 × 10^−5^	AT2G34430	3	chlorophyll a-b binding protein of LHCII type 1	GO_BP,GSEA_CC
*Zm00001d044399*	1341.4	−3.92	0.000709	AT2G34430	3	chlorophyll a-b binding protein 2	GO_BP,GSEA_CC
*Zm00001d011285*	6415.27	−4.45	3.32 × 10^−7^	AT2G34420	8	light harvesting chlorophyll a/b binding protein 1	GO_BP,GSEA_CC
*Zm00001d044401*	1802.37	−4.75	2.21 × 10^−7^	AT2G34430	3	chlorophyll a-b binding protein of LHCII type 1	GO_BP,GSEA_CC
*Zm00001d044402*	2213.89	−4.75	1.11 × 10^−6^	AT2G34430	3	chlorophyll a-b binding protein of LHCII type 1	GO_BP,GSEA_CC

GO_BP: GO biological process. GSEA_CC: Gene set enrichment analysis cell component.

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
