# Peer review of "Dissecting the Regulatory Network of Leaf Premature Senescence in Maize (Zea mays L.) Using Transcriptome Analysis of ZmELS5 Mutant"

_genes, 2019, doi:10.3390/genes10110944_

Round 1
Reviewer 1 Report
In the manuscript by Chai et al., the authors compared the transcriptome between a leaf premature senescence mutant and wild type. The authors identified a couple of thousands up- or down-regulated genes between the two lines. They also performed GO analysis and built regulatory networks. This work used unique plant materials, which show early senescence, to explore the mechanism of leaf senescence. It contains interesting information that will be useful for both plant scientists and breeders. However, I still have some minor suggestions/questions for this manuscript.
What is the size of the genome region from the donor parent? It is important because many of the up- or down-regulated genes might be affected by other genes in this region, not directly associated with the gene that induces early senescence. I assume the authors already have a mapping region for the early senescence mutant. But still premature to publish those results. I am wondering are there any of the differentially expressed genes in the mapping region? Are there any of the Pathogenesis-related (PR) genes differentially expressed between the mutants and the wild type? If there are any, the authors should highlight and discuss them. The font size in most of the figures is too small. The authors should increase it to make sure readers see the text clearly.
Author Response
Thank very much for your critical comments for our manuscript. We have revised the manuscript carefully according to your suggestions. The major revisions are as following:
Response to Reviewer #1
What is the size of the genome region from the donor parent? It is important because many of the up- or down-regulated genes might be affected by other genes in this region, not directly associated with the gene that induces early senescence. I assume the authors already have a mapping region for the early senescence mutant. But still premature to publish those results. I am wondering are there any of the differentially expressed genes in the mapping region?
In our previous study, the candidate gene of ZmELS5 mutant senescence was mapped to the chromosomal region Chr8: 86964414-92548025. In this region, there several differently expressed genes distributed, such as Zm00001d009945, Zm00001d009910, Zm00001d009903, Zm00001d009928, and Zm00001d009936. We have added a description in the discussion section.
Are there any of the Pathogenesis-related (PR) genes differentially expressed between the mutants and the wild type? If there are any, the authors should highlight and discuss them.
Of the identified differentially expressed gene, there were several Pathogenesis-related (PR) genes. And, we have a discussion in corresponding section.
The font size in most of the figures is too small. The authors should increase it to make sure readers see the text clearly.
We have re-made Fig. 3-11 to make sure readers see the text clearly.
Reviewer 2 Report
This work provides an original manuscript (MS) on gene leaf senescence network in a maize mutant.
Nevertheless, the MS needs some clarifications in some methodology points, such as qPCR. Why Actine Gene is used? Did you try any other reference genes (RG) to give evidence that this the most stable for gene expression measurments? Would you please provide these data in case you have tested another RGs?
With respect to Language Editing, there are many words would be improved in its writing. FOr example, when authors write that leaf senescence is a complicated process, it should be better to say that it is a complex process.
I suggest improving English style writing.
Additionally, when discussing your results on TFs associated to leaf senescence process, you can revise the lately published review to enrich your discussion (https://www.mdpi.com/2223-7747/8/10/411/htm)
Author Response
Thank very much for your critical comments for our manuscript. We have revised the manuscript carefully according to your suggestions. The major revisions are as following:
Response to Reviewer #2
This work provides an original manuscript (MS) on gene leaf senescence network in a maize mutant. Nevertheless, the MS needs some clarifications in some methodology points, such as qPCR. Why Actine Gene is used? Did you try any other reference genes (RG) to give evidence that this the most stable for gene expression measurements? Would you please provide these data in case you have tested another RGs?
In qRT-PCR analysis, we used Actin gene as the only reference gene. Actin gene is a common reference gene in plant, and it has been used in many studies. In this work, the results of qRT-PCR were highly in consistent the results of transcriptome sequencing. This indicated that Actin gene is an idea reference gene in maize.
With respect to Language Editing, there are many words would be improved in its writing. For example, when authors write that leaf senescence is a complicated process, it should be better to say that it is a complex process. I suggest improving English style writing.
We have revised the English writing carefully.
Additionally, when discussing your results on TFs associated to leaf senescence process, you can revise the lately published review to enrich your discussion (https://www.mdpi.com/2223-7747/8/10/411/htm).
According to the published review that the Reviewer provided, we have rewrote the corresponding section of the discussion.